# Biallelic Optic Atrophy 1 (*OPA1*) Related Disorder—Case Report and Literature Review

**DOI:** 10.3390/genes13061005

**Published:** 2022-06-02

**Authors:** Bayan Al Othman, Jia Ern Ong, Alina V. Dumitrescu

**Affiliations:** 1Flaum Eye Institute, University of Rochester Medical Center, Rochester, NY 14642, USA; bayan_alothman@outlook.com; 2College of Osteopathic Medicine, Des Moines University, Des Moines, IA 50312, USA; jiaern.ong@dmu.edu; 3Pediatric Ophthalmology and Strabismus, Inherited Eye Disease, University of Iowa Hospitals and Clinics, Iowa, IA 52242, USA

**Keywords:** Behr disease, optic neuropathy, biallelic *OPA1*

## Abstract

Dominant optic atrophy (DOA), MIM # 605290, is the most common hereditary optic neuropathy inherited in an autosomal dominant pattern. Clinically, it presents a progressive decrease in vision, central visual field defects, and retinal ganglion cell loss. A biallelic mode of inheritance causes syndromic DOA or Behr phenotype, MIM # 605290. This case report details a family with Biallelic Optic Atrophy 1 (*OPA1*). The proband is a child with a severe phenotype and two variants in the *OPA1* gene. He presented with congenital nystagmus, progressive vision loss, and optic atrophy, as well as progressive ataxia, and was found to have two likely pathogenic variants in his *OPA1* gene: c.2287del (p.Ser763Valfs*15) maternally inherited and c.1311A>G (p.lIle437Met) paternally inherited. The first variant is predicted to be pathogenic and likely to cause DOA. In contrast, the second is considered asymptomatic by itself but has been reported in patients with DOA phenotype and is presumed to act as a phenotypic modifier. On follow-up, he developed profound vision impairment, intractable seizures, and metabolic strokes. A literature review of reported biallelic *OPA1*-related Behr syndrome was performed. Twenty-one cases have been previously reported. All share an early-onset, severe ocular phenotype and systemic features, which seem to be the hallmark of the disease.

## 1. Introduction

Dominant optic atrophy (DOA) MIM # 605290 is the most common hereditary optic neuropathy and is typically inherited in an autosomal dominant pattern with a prevalence of 1:12,000 to 1:50,000 [1]. It is most commonly associated with pathogenic variants in the optic atrophy 1 gene (*OPA1*) [2], which encodes mitochondrial dynamin-related GTPase responsible for mitochondrial membrane stabilization. The OPA 1 protein is shown to play an essential role in mitochondrial fusion, control of cristae morphology and apoptosis [3], as well as maintenance of membrane potential, oxidative phosphorylation, stability, and maintenance of mtDNA [4]. Clinically, DOA is characterized by progressive visual field and color vision deficits, a distinctive pattern of temporal pallor of the optic nerves, and retinal ganglion cell loss with a variable effect on visual acuity. It is usually diagnosed during early school age for patients referred with reading difficulties, but diagnosis can be delayed for several decades in patients with mild symptoms. The vision loss is usually insidious and slowly progressive, although rapid vision loss can occur. The patients typically have a moderate, irreversible visual impairment; although severe and very mild, vision loss has been reported [1]. More than 500 individual variants in *OPA1* have been reported in association with DOA [5]. In 20% of cases, patients have been described as “DOA plus,” presenting with additional systemic manifestations including neuromuscular features like ataxia, peripheral neuropathy, mitochondrial myopathy, and various degrees of sensorineural hearing loss as well as chronic progressive external oph thalmoplegia, nystagmus and seizures [6]. One study found that about 30% of the time, family members carrying the same *OPA1* variant manifest variable phenotype (both DOA and DOA+) [6].

In 1909, Behr presented a series of “complicated familial optic atrophy with childhood-onset,” including neurological abnormalities like ataxia, cranial nerve palsies, ophthalmoplegia, spastic paraparesis, nystagmus, peripheral neuropathy, and variable degrees of intellectual disabilities in addition to vision loss [7]. This combination of clinical findings is currently known as Behr syndrome, MIM # 605290. The optic atrophy associated with Behr syndrome can be progressive, but other neurological abnormalities usually progress and become more prominent during the second and third decades of life. Most of the cases are sporadic or inherited in an autosomal recessive pattern, although autosomal dominant inheritance was reported in some families [8]. This syndromic optic atrophy was linked initially to autosomal recessive *OPA3* pathogenic variants with elevated urinary levels of 3-methylglutaconic acid and 3-methylglutaric acid [9]—a subtype that was also known as Costeff syndrome [10,11] but recently a biallelic *OPA1* inheritance has been suggested to cause Behr phenotype. Some of the patients reported as biallelic inheritance carry one known *OPA1* mutation and a second variant (considered nonpathogenic in itself) acting as a phenotypic modifier [12]. This paper presents a family with such biallelic *OPA1* inheritance, describes phenotypic and genotypic details and long-term follow-up and reviews the literature for similar reports.

## 2. Materials and Methods

Case report of a family in which the son presented with likely biallelic *OPA1* Behr syndrome. Data extracted from the records included clinical exams, ancillary testing, genetic testing results, and additional morbidity. Genetic testing—Optic atrophy sequencing panel, was performed by Prevention Genetics laboratories in Marshfield, Wisconsin. The panel included 16 different genes known to be associated with optic atrophy (*ACO2*, *AUH*, *C12orf65*, *CISD2*, *MFN2*, *MTPAP*, *NDUFS1*, *NR2F1*, *OPA1*, *OPA3*, *POLG*, *SLC24A1*, *SBG7*, *TIMM8A*, *TMEM126A*, *WFS1*) and also included copy number variants (deletions and duplications) testing. A combination of next-generation sequencing (NGS) and Sanger sequencing technology was used to cover the entire coding regions of the listed genes plus approximately 20 bases of non-coding DNA flanking each exon. Genomic DNA was extracted from the Patient’s blood. Patient DNA corresponding to these regions was captured for NGS using an optimized set of DNA hybridization probes. Captured DNA was sequenced using Illumina’s Reversible Dye Terminator (RDT) platform (Illumina, San Diego, CA, USA). Minimum NGS coverage is more than 20 times for all exons and +/− 10 base pairs of intronic flank and more than 10 times from 11–20 base pairs of intronic flank. All regions with coverage that did not meet this threshold were backfilled with Sanger sequencing. All pathogenic, undocumented, and suspected NGS variant calls were confirmed by Sanger sequencing. For Sanger sequencing, polymerase chain reaction (PCR) was used to amplify targeted regions. After purification of the PCR products, cycle sequencing was carried out using the ABI Big Dye Terminator v.3.1 kit. All PCR products were resolved by electrophoresis on an ABI 3730 × capillary sequencer.

Written Consent for publication and use of clinical data and photos of this case was obtained from the mother and father for themselves and their son. 

The literature review was performed by searching PubMed for the terms Behr syndrome, biallelic *OPA1*, and syndromic optic atrophy.

## 3. Results

### 3.1. Case Description 

A 17-months-old Caucasian male was referred to the pediatric ophthalmology clinic to evaluate for congenital nystagmus, decreased vision, delayed motor skills and delayed walking milestones. He was reported as otherwise healthy and had a history of being delivered via assisted vaginal delivery at 35 6/7 weeks to a 28-year-old female G1, P1 with a birth weight of 2480 g. His postnatal course was uneventful. Ancestry includes German/Scottish/English/Mexican ancestry. No history of consanguinity. 

On ocular examination, the patient’s vision was central, unsteady, and maintained (CUSM) in the right (OD) and the left (OS) eyes. Visual acuity measurements by Teller acuity cards (TAC) were 2.4 cy/cm (20/260) in both eyes (OU) at 55 cm. He had horizontal, large amplitude, pendular nystagmus with a rotary component more obvious in upgaze. The slit-lamp exam was unremarkable. Fundus examination of both eyes revealed +2 optic disc pallor with a large area of retinal pigment epithelium (RPE) atrophy surrounding the optic nerves and extending to the temporal macula (Figure 1a,b).

The patient had no strabismus. Cycloplegic refraction showed low myopia and astigmatism. Sedated full-field electroretinogram (ffERG) was virtually normal, showing low normal dark-adapted ERG with normal light-adapted ERG amplitudes (Figure 2a). Magnetic resonance imaging (MRI) of the brain with and without contrast at the time of presentation was normal. Awake, flash visual evoked potential (fVEP) was recordable and showed low amplitudes and delayed latency OU, OD worse than OS. (Figure 2b). The optical coherence tomography (OCT) showed formed fovea and thinning of the retinal nerve fiber layer (RNFL) OU. (Figure 3a,b). All of the above findings were highly suggestive of optic nerve pathology.

The patient was referred to the pediatric neurology clinic to evaluate ataxia and developmental delay. His neurological exam was significant for grossly normal sensation, strength, and muscle tone; hypoactive deep tendon reflexes; and ataxia affecting limb function and gait. The hearing was normal.

Family history includes polycystic ovary syndrome and a history of infertility of the mother. The maternal side of the family has a significant history of cancer, including Non-Hodgkin’s lymphoma in a maternal great-aunt, maternal ovarian cancer in a great-great-aunt, brain cancer in a maternal great-great-uncle, and leukemia in a great-great-great-grandmother and great uncle. The mother is a known carrier of the ataxia-telangiectasia mutated (ATM) gene through genetic testing for familial cancer. The father has a history of complex partial and grand mal seizures, well-controlled on oral carbamazepine. He has a mild speech impairment that has been attributed to the seizures. There was no family history of visual impairment, nystagmus, or ataxia.

The clinical diagnosis was DOA plus–Behr syndrome, and genetic testing was discussed and performed after pretesting genetic counseling.

### 3.2. Genetic Testing

Chromosome microarray analysis of the proband was normal. Additional genetic testing, using an “optic atrophy sequencing panel,” was positive for two variants in his *OPA1* gene: c.2287del (p.Ser763Valfs*15) NM_130837.2 and c.1311A>G (p.lIle437Met) NM_130837.2. He is a carrier for the same heterozygous maternal mutation in ATM gene, sequencing-c.2284_2285del. Testing was negative for other variants and deletions or duplications.

The variant c.2287del (maternally inherited) is predicted to result in a translational frameshift and premature protein termination (p.Ser763Valfs*15). The same variant is alternatively denoted as NM_015560.2: c.2122del (p.Ser708Valfs*15. The functional analysis of the effect of this mutation on the amount of protein was not performed. This variant was previously reported as being pathogenic and likely to cause DOA [13]. The second sequence variant, designated c.1311A>G (paternally inherited), is predicted to result in the amino acid substitution p.IIe437Met. This variant, which is also commonly referred to as c.1146A>G (p.IIe382Met), is considered asymptomatic by itself but has been reported in at least 16 other patients with DOA phenotype [14,15,16,17] and is presumed to act as a phenotypic modifier [18]. This variant has been reported in public allele frequency databases with an overall allele frequency of 0.06%. The amino acid residue p.Ile437 of the OPA1 protein has been highly conserved during evolution. The amino acid substitution prediction program PolyPhen2, SIFT, and MutationTasteor predict the p.Ile437Met change to be damaging, but it has been associated only with asymptomatic/very mild DOA. It has been shown that on its own, this variant did not perturb mitochondrial morphology and mtDNA content [19]. This variant has been reported in a patient affected by autosomal dominant bilateral optic atrophy [20] in the heterozygous state [21,22]. Several previous reports indicated that c.1311A>G might be a mild pathogenic variant that might contribute to increased severity of the disease when present in the compound heterozygous state with a known pathogenic variant [12,23].

**Figure 2 genes-13-01005-f002:**
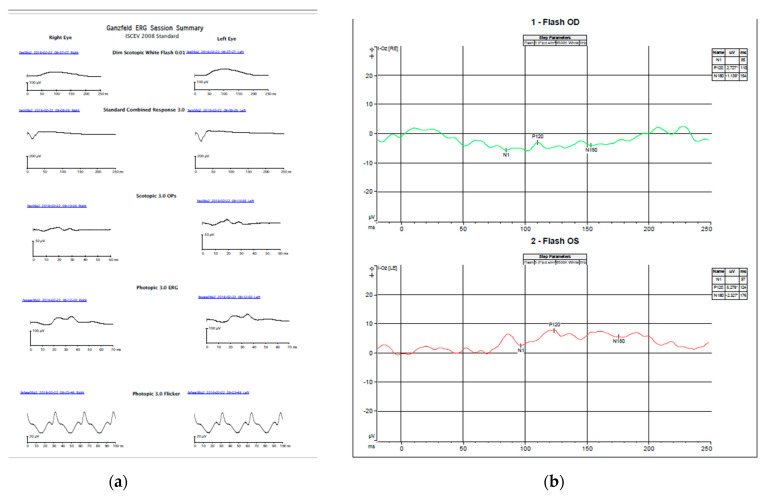
Electrophysiology of the child (**a**) sedated full-field ERG using Burien Allen electrodes and the Vera system. Both dark and light-adapted waveforms and amplitudes were normal compared with our standards for sedated ERG (**b**) Awake flash VEP performed using a Dyagnosis system. Very low, almost non-recordable amplitudes bilaterally, right eye worse than left. Pick amplitudes for the P120 waves are 2.27 mV right eye and 5.27 mV left eye. Normal amplitudes are over 6 mV [24].

**Figure 3 genes-13-01005-f003:**
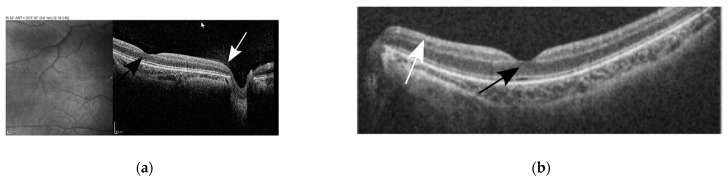
Optical coherence tomography (OCT) of the right (**a**) and left (**b**) macula of the child shows formed fovea in both eyes (black arrows) and a thin retinal neuron fiber layer (white arrows).

**Table 1 genes-13-01005-t001:** Summary of previous cases of patients with compound heterozygous optic atrophy 1 (*OPA1*) pathogenic variants mo—months; y—years; M—male; F—female.

Reference	Age of Onset	Gender	Mutation/Amino Acid Change		Visual Acuity	Optic Atrophy	Ataxia	Peripheral Neuropathy	Seizures	VEP	Brain MRI
			Allele 1	Allele 2							
The current report	Birth	M	c.1311A>G/ p.lIle437Met	c.2287del/ p.Ser763Valfs*15	0.05/0.1	Yes	Yes	No	status epilepticus and intractable seizures	Low amplitude	Multiple metabolic strokes in his left thalamus, left occipital lobe, and left frontal lobe
Bonneau et al. [23]	42 mo	F	c.1146A>G/ p.Ile382Met	c.2470C>T/ p.Arg824 *	0.01/0.01	Yes	Yes	Yes	Not reported	Abnormal	Normal
Bonneau et al. [23]	12 mo	F	c.1204G>A/ p.Val402Met	c.2708-2711del4/ p.Val903Glyfs *	unknown	Yes	Yes	Yes	Not reported	Abnormal	Cerebellar atrophy
Bonneau et al. [23]	18 mo	M	c.1146A>G/ p.Ile382Met	c.1669C>T/ p.Arg557 *	0.01/0.01	Yes	Yes	Yes	Not reported	Abnormal	Cerebellar atrophy
Bonneau et al. [23]	3 y	M	c.1146A>G/ p.Ile382Met	c.1459G>A/ p.Glu487Lys	Blind	Yes	Yes	Yes	Not reported	Abnormal	Cerebellar atrophy. Hypoplasia of the optic chiasm and optic nerves
Schaaf et al. [25]	12 mo	M	c.1146A>G/ p.Ile382Met	c.2708-2711del4/ p.Val903Glyfs *	Reduced severely	Severe, diffuse	Yes	Yes	No	unknown	Periventricualr leucomalacia
Schaaf et al. [25]	6 mo	F	c.1146A>G/ p.Ile382Met	c.2708-2711del4/ p.Val903Glyfs *	unknown	Severe, diffuse	Yes	Yes	No	unknown	Not performed
Pesch et al. [26]	Childhood	F	c.808G>A/ p.Glu270Lys	c.868C>T/ p.Arg290Trp	0.2/0.2	Diffuse pallor	Not reported	Yes	No	Abnormal—delayed latencies	unknown
Lee et al. [27]	12 mo	M	c.2714G>A/ p.Arg905Gln	c.1857-1858delinsT/ p.Leu620fs*13	Blind. Congenital cataract	Diffuse pallor	Yes	Yes	No	Absent	Normal
Yu-Wai-Man et al. [6]	unknown	M	c.768C>G/ p.Ser256Arg	c.854A>G/ p.Gln285Arg	Unk	Yes	Yes	Yes	Not reported	unknown	unknown
Yu-Wai-Man et al. [6]	unknown	F	c.768C>G/ p.Ser256Arg	c.854A>G/ p.Gln285Arg	Unk	Yes	Yes	Yes	Not reported	unknown	unknown
Carelli et al. [28]	1 y	M	c.1311A>G/ p.lIle437Met	c.1705+1G>T	LP	Yes	Yes	Yes	No	Delayed	Normal
Bonifert et al. [12]	2 y	M	c.1311A>G/ p.lIle437Met	c.610+364G>A	0.02	Yes	Yes	Yes	No	unknown	Cerebellar atrophy and thinning of the cervical spinal cord
Bonifert et al. [12]	2 y	M	c.1311A>G/ p.lIle437Met	c.610+364G>A	0.02	Yes	Yes	Yes	No	unknown	Cerebellar atrophy and thinning of the cervical spinal cord
Bonifert et al. [12]	2 y	F	c.1311A>G/ p.lIle437Met	c.610+364G>A	0.02	Yes	Yes	Yes	No	unknown	Cerebellar atrophy and thinning of the cervical spinal cord
Bonifert et al. [12]	1 y	F	c.1311A>G/ p.lIle437Met	c.1316_1317insA/ p.N440Kfs*14	unknown	Yes	Yes	Yes	No	unknown	Unknown
Nasca et al. [15]	1 mo	M	c.1311A>G/ p.lIle437Met	c.190_194del/ p.Ser64Asnfs*7	unknown	Yes	Yes	Yes	Present	unknown	Cerebellar atrophy, edema in the occipito-parietal cortical areas
Nasca et al. [15]	4-6 y	F	c.1311A>G/ p.lIle437Met	c.2962G>T/ p.Val988Phe	unknown	Yes	Yes	Yes	No	Abnormal - optic nerve dysfunction	Optic nerves and chiasm atrophy
Nasca et al. [15]	4 y	F	c.1180G>A/ p.Ala394Thr	c.1180G>A/ p.Ala394Thr	unknown	Unk	Yes	Yes	No	Abnormal - optic nerve dysfunction	Abnormal hyperintensities of the periventricular and centrum semiovale white matter areas, mild global cerebellar atrophy
Zerem et al. [14]	18 mo	F	c.1146A>G/ p.Ile382Met	c.1963_1964dupAT/ p.Lys656fs	0.008/0.004	Yes	Yes	Yes	Not reported	Low amplitude responses	Atrophy of the optic nerves, mild central and deep white matter multifocal T2 hyperintensities. Right parietal acute infarct. Metabolic stroke
Rubegni et al. [29]	5 y	F	c.1180G>A/ p.Ala394Thr	c.1180G>A/ p.Ala394Thr	unknown	Optic disc pallor	Yes	Yes	No	unknown	Progressive cerebellar involvement accompanied by basal ganglia hyperintensities
Rubegni et al. [29]	8 mo	M	c.2779-2A>C	c.2809C>T/ p.Arg937Cys	Blindness	Severe optic atrophy	Yes	Yes	No	Increase latency and reduced amplitude	Progressive cerebellar involvement accompanied by basal ganglia hyperintensities

### 3.3. Parental Examination

The mother’s initial clinical exam showed best-corrected visual acuity (BCVA) of 20/20 OD and OS., normal reactive and symmetrical pupils with no relative afferent pupillary defects (RAPD), normal intraocular pressure (IOP) OU, normal 30-2 Humphrey visual fields OU, normal color vision OU, unremarkable anterior segment exam OU. and slight temporal pallor of both optic nerves (Figure 4a,b), right eye more than left, consistent with clinically mild *OPA1* disease. Optic nerves OCT showed mild thinning of the RNFL and ganglion cell layers (GCL) OU (Figure 5a,b). She has an abnormal pattern ERG showing normal P50 amplitudes indicating normal photoreceptor function and decreased N95 amplitudes indicating ganglion cell dysfunction (Figure 6a,b) and a slightly abnormal pattern VEP (Figure 7a,b), both confirmatory of mild optic atrophy phenotype. The mother’s neurologic exam was normal, including hearing.

The father’s clinical exam showed BCVA of 20/20 in either eye, normal-reactive and symmetrical pupils with no RAPD, normal IOP, normal Goldman visual field OU, normal color vision OU, healthy-looking optic discs OU, and normal fVEP and OCT OU. These findings are consistent with no evidence of disease at this point. The father’s neurologic exam was positive only for mild speech abnormalities, with normal hearing. The complete family pedigree is presented in Figure 8.

### 3.4. Management and Disease Progression

The follow-up for this case was six years to age 8.

Management generally consisted of periodic eye exams and maximizing visual potential with correction of refractive errors (low myopia and moderate astigmatism OU). Compliance with the glasses was fair. Visual rehabilitation and low vision services were provided to the patient and consisted of magnifying devices and text-to-speech technology. He started learning braille and the use of tactile and auditory approaches for learning. Also, he began to use a cane for mobility.

The patient’s visual acuity was variable but progressively declined on follow-up exams, with the right eye more than the left. At age 5.5, he tested 20/400 OD and 20/200 OS with an eccentric fixation on LEA symbols. Soon after he turned 6, a marked decline in his visual function was noted by parents and educators. The examination showed his visual acuity of hand motion of one foot and 0.32 cy/cm and 38 cm (20/2700) on TAC. He was unable to perform visual fields or color vision testing. Fundus examination showed progressive optic disc pallor to +4 OU. Flash VEP became non-recordable. Over time, the patient developed large-angle exotropia of more than 50 prism diopters. At this point, parents are not interested in surgical correction.

Initially, he passed the hearing screening on two occasions; however, at age 4, he started showing mild hearing impairment in the left ear, which further progressed to moderate bilateral hearing loss.

He was a bright and fun boy. His follow-up neurologic examinations showed symmetrically hypoactive deep tendon reflexes, intention tremor, dysmetria, wide-based, ataxic gait with pronation of the ankles with both walking and standing, and no evidence of myopathy or neuropathy. Parents reported worse and more unsteady walking when he was tired or ill. He developed some repetitive staring spells that are investigated as possible absence seizures. Electroencephalogram (EEG) was negative for seizures but showed abnormal waveforms and slowing in bilateral occipital lobes. Suddenly, at the age of 6.3 years, he presented with status epilepticus in the setting of acute Haemophilus influenza infection. He was admitted to the pediatric intensive care unit (PICU) and was treated with escalating medications to treat the status epilepticus, which finally resolved after achieving burst-suppression with pentobarbital. He underwent a workup with a lumbar puncture to assess for infection or autoimmune encephalitis; studies of this spinal fluid were ultimately unremarkable. He was treated with intravenous immunoglobulin and was weaned off pentobarbital to levetiracetam, lacosamide, ketogenic diet, and levocarnitine. Initial MRI showed diffusion restriction involving the left frontal cortex, which resolved on a repeat MRI 3 days later. With the recovery from the seizure activity, it was apparent that the patient had lost several of his previous skills, including his ability to walk, talk, and eat without assistance. He was discharged home with the plan for intensive rehabilitation. Unfortunately, he returned to the hospital twice with status epilepticus super refractory to treatment and developed intractable epilepsy and multiple metabolic strokes in his left thalamus, left occipital lobe, and left frontal lobe. After his last admission, he can only move one upper extremity and is G tube dependent. His visual acuity is light perception.

Ophthalmology follow-up on the patient’s mother (over the same interval of time) showed mildly decreased visual acuity to (20/25), not improving with refraction, stable (abnormal) optic nerve OCT and abnormal pattern ERG and VEP.

Father’s examination was stable and normal.

## 4. Discussion and Literature Review

*OPA1* pathogenic variants can profoundly affect vision. Patients can have variable presentation and prognosis, resulting in an extensive range of visual and neurological impairments. The clinical spectrum of these pathogenic variants has expanded over the last 15 years to include biallelic inheritance, which has been suggested to cause more severe disease in affected individuals, [30] and to include cases of patients with extraocular dysfunction preceding or presenting in the absence of optic neuropathy [31].

We present in this manuscript a family (mother, father, and the son) who carry variants in *OPA1* gene, with the son (who carries both variants) having a severe phenotype including significant progressive visual impairment and neurological abnormality is, the mother (who carries a known DOA pathogenic variance) having mild phenotype, and the father (who carries the likely mild, phenotypic modifier) being unaffected at this point. Both parents were diagnosed after their son was determined to have Behr syndrome.

Although heterogeneous, most patients with hereditary optic neuropathy (>60%) harbor pathogenic variants within *OPA1*, and ~3% have *OPA3* pathogenic variants. Marelli et al. previously described two brothers with classic Behr syndrome who have a single heterozygous pathogenic *OPA1* pathogenic variant (c.1652G4A, p.Cys551Tyr) within the catalytic GTPase domain [20,32]. Other reports showed DOA plus phenotype with a single heterozygous pathogenic *OPA1* variant, proving variability [6].

The case presented above and the literature review in this paper focused on biallelic *OPA1* inheritance with documented genetic testing (Table 1). Bonneau et al. published a case series of four patients with compound heterozygous *OPA1* pathogenic variants with the co-occurrence of a missense GTPase mutation and a truncating nonsense mutation. All four children have typical clinical features of progressive optic neuropathy with ataxia, spasticity, and peripheral neuropathy [23]. Bonifert et al. showed for the 1st time evidence of a modifier effect of *OPA1* variant c.1311A>G/p.Ile437Met (the same as the presumably unaffected father in our pedigree) as the cause of optic atrophy plus phenotypes by a combined mutational effect [12]. Another DOA plus family with compound heterozygous missense GTPase *OPA1* pathogenic variants (c.1146A4G, p.Ile382Met) has been reported [25]. These cases suggest that this particular variant is highly penetrant for the neurological “plus” features with a more potent deleterious impact when inherited biallelic with another pathogenic variant, although it is not pathogenic on its own or even in homozygous status as showed by Bonifert et al.

Lee et al. reported a 10-year-old patient with Behr syndrome presenting with early-onset severe 30optic atrophy, peripheral neuropathy, ataxia, and congenital cataracts, which has a pair of compound heterozygous pathogenic variants: p.L620fs*13 (c.1857–1858delinsT) and p.R905Q (c.G2714A) [27].

Pesch et al. also described an adult patient with DOA, a compound heterozygote for two *OPA1* missense pathogenic variants [26]. A literature search found few other reports of patients with compound heterozygous OPA1 pathogenic (21 individual cases plus ours) [6,12,14,15,20,28]. Table 1 summarizes all the previously reported cases and their clinical features. These cases share common extraocular features, including sensorineural deafness, ataxia, peripheral neuropathy, and myopathy. Other rare manifestations have been reported, such as spastic paraplegia, multiple sclerosis-like syndrome [6], metabolic stroke [14], syndromic parkinsonism and dementia [28], and Behr-like syndrome [32]. Most of these dysfunctions affect the central, peripheral, and autonomous nervous systems, highlighting the close relationship between functions of the *OPA1* gene and neuronal physiology [30]. The metabolic strokes and neurologic impairment are due to the mitochondria’s impaired response to oxidative stress. Most of the cases show severe visual impairment and early onset. A report from Liao et al. showed increased mitochondrial fragmentation and recycling due to decreased levels of OPA 1 protein which is essential for mitochondrial fusion [33]. Most patients are wheelchair-bounded in their twenties due to progressive myopathy and peripheral neuropathy. Approximately half of the patients described here had nystagmus (congenital or acquired) and virtually all had ataxia.

As shown in Table 1, the case presented here has a particularly severe ocular and neurologic phenotype. Of the 22 cases (21 + our proband), there are only two frameshift mutations—our proband and the patient described by Zerem et al. Both patients have the same mutation on their 2nd allele and have in common the development of metabolic strokes. On the database of all known *OPA1* pathogenic variants, 22% are reported as frameshift mutations; however, a phenotype-genotype correlation was not established [5]. On the other hand, the Ile437Met variant is present in 12 of the 44 total alleles. Alleles Ile437Met and Ile382Met most likely denote the same mutation but are annotated based on different reference sequences.

Behr syndrome has been reported to be caused by pathogenic variants in both copies of the *OPA1* gene, the *OPA3* gene, or the *C12ORF65* gene. Our Patient’s symptoms and the fact that he has a variant in both copies of the *OPA1* gene are consistent with Behr syndrome, which can also be considered a more severe, early-onset form of *OPA1*.

Behr syndrome due to biallelic pathogenic variants in *OPA1* is exceptionally rare, with only 21 reported cases in the English language medical literature. A common clinical feature of these cases is a more severe overall phenotype and earlier onset (early childhood), and poor visual and neurologic prognosis. There is currently no specific treatment for this disease. Supplements have been suggested but have not proven beneficial [34].

## 5. Conclusions

While this report does not provide a new genetic finding, it reminds us how rare Behr syndrome with biallelic pathogenic variants in *OPA1* is. The case report details the clinical course of proband over six years in the clinical course of his parents. While the mother had only mild disease (able to be identified only on OCT and electrophysiology), the child had a severe phenotype and a very dramatic disease progression. Early-onset visual impairment with Behr syndrome phenotype should be considered suspicious for biallelic *OPA1* variants. Biallelic inherited variants show a strong additive effect on the phenotype and tend to be much more severe when a pathogenic *OPA1* variant is combined with a second mutation. The prognosis for vision and neurologic impairment is guarded and should be included in the counseling for the patient and family. Phenotypic features are essential for diagnosis; however, genetic testing, including parental testing, is the final step in having a correct and complete diagnosis.

## Figures and Tables

**Figure 1 genes-13-01005-f001:**
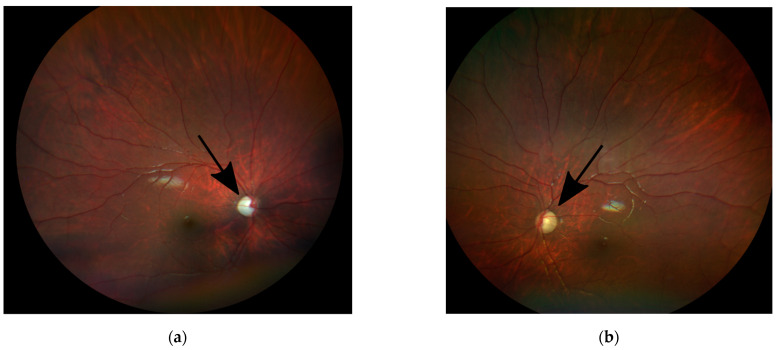
Fundus photos of the right (**a**) and left (**b**) eyes of the child showing diffuse optic disc pallor in both eyes (black arrows).

**Figure 4 genes-13-01005-f004:**
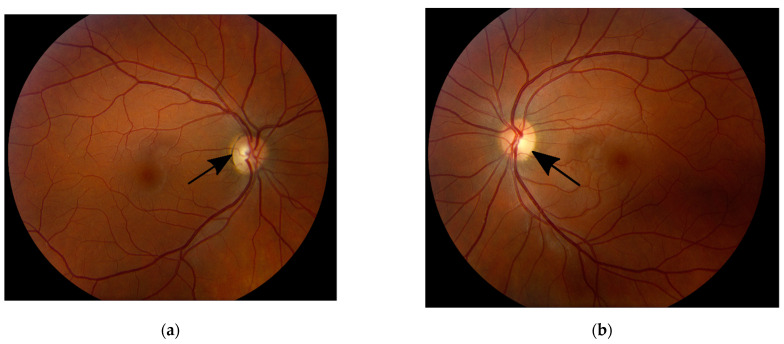
Fundus photos of the right (**a**) and left (**b**) eyes of the mother showing mild temporal optic disc pallor in both eyes (black arrows).

**Figure 5 genes-13-01005-f005:**
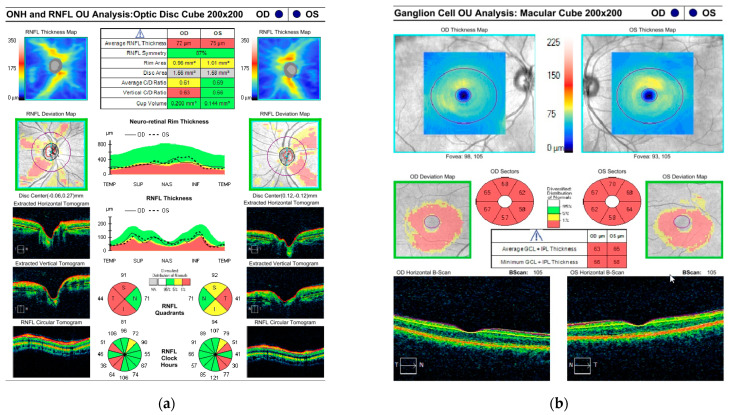
Optical coherence tomography (OCT) of the retinal nerve fiber layer (RNFL) (**a**) and ganglion cell layer (GCL) (**b**) of the mother. Red color represents thinning below normal values. Yellow color represents a moderate thinning. Green color represents normal values.

**Figure 6 genes-13-01005-f006:**
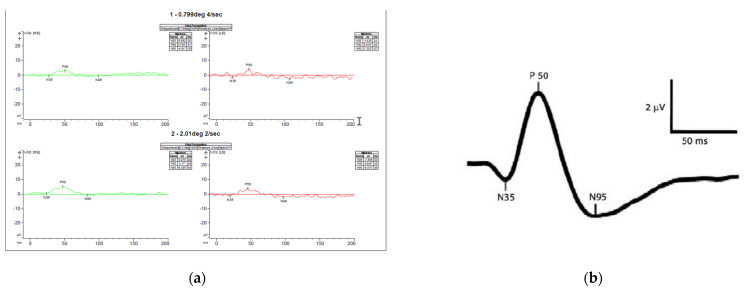
Pattern ERG (**a**) recordings of Patient’s mother showing normal P50 amplitudes indicating normal photoreceptor function and decreased N95 amplitudes (not below the baseline) indicating ganglion cell dysfunction. (**b**) normal pattern ERG waveforms-the P50-wave is the initial positive deflection originating from RGCs and outer retinal photoreceptor cells, namely the macular cones. The N95-wave is the negative deflection following the P50-wave that originates from the inner retina. This wave component reflects the RGC function. The asterisks * means the values are relative to the baseline (flat line) and that is why some are positive (p) and other negative (n).

**Figure 7 genes-13-01005-f007:**
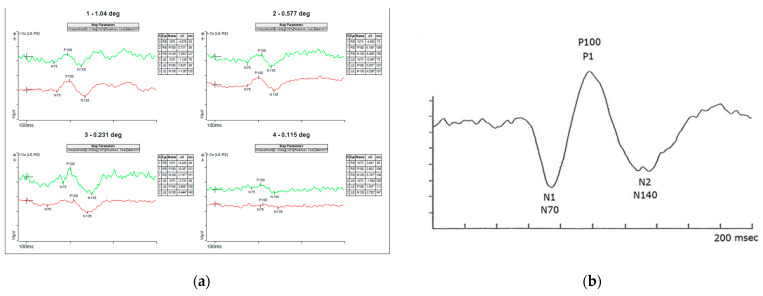
Pattern VEP (**a**) recordings of Patient’s mother showing slightly decreased amplitudes more significant (**b**) normal pattern VEP waveforms showing the amplitude and latency to N70, P100 and N155 waves. The asterisks * means the values are relative to the baseline (flat line) and that is why some are positive (p) and other negative (n).

**Figure 8 genes-13-01005-f008:**
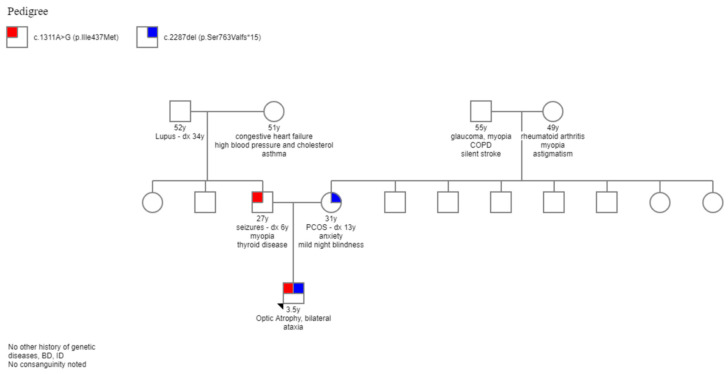
Complete family pedigree. Red colored is the paternally inherited variant and blue color is the maternally inherited variant.

## Data Availability

Not applicable.

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
