# Peer review of "Biallelic Optic Atrophy 1 (OPA1) Related Disorder—Case Report and Literature Review"

_genes, 2022, doi:10.3390/genes13061005_

Round 1

Reviewer 1 Report

This is a well-considered, detailed and interesting report with comprehensive phenotyping. A few minor points:

  • The manuscript needs some attention to grammar and English throughout- there are frequently missed words such as ‘it’ and ‘the’. For example, in the abstract ‘clinically presents’ should be ‘clinically it presents’ and line 206-7 ‘Over time, the patient developed large-angle exotropia of more than 50 prism diopters. At this point, parents are not interested in surgical correction.’ Should read ‘Over time, the patient developed a large-angle exotropia of more than 50 prism diopters. At this point, his parents were not interested in surgical correction.’
  • Consider adding MIM numbers to the diseases discussed
  • Genetic testing ‘The panel included 16 different genes known to be associated with optic atrophy 138 (ACO2, AUH, C12orf65, CISD2, MFN2, MTPAP, NDUFS1, NR2F1, OPA1, OPA3, POLG, 139 SLC24A1, SBG7, TIMM8A, TMEM126A, WFS1) and also included copy number variants 140 (deletions and duplications) testing.’ This paragraph should be in section 2 methods. It would also be useful to expand on the methods used in the panel test- how was the DNA prepared and sequenced. Which reference genome was used, how was the CNV analysis performed?
  • Which Genbank accession number is being used for your OPA1 variants?
  • Figure 1 perhaps zoom in on the discs
  • Figure 2 needs to be larger to see the ERG data properly
  • Figure 4 maternal imaging, does the RNFL and GCL really show only mild disease? Whilst her visual function is still good there is significant thinning shown

Author Response

Thank you for reviewing our manuscript. Attached is a response to your feedback.

Reviewer 2 Report

In this work, Dumitrescu and colleagues present a case report of a biallelic Optic Atrophy (OPA1) related disorder and compared it with other biallelic OPA1 disorders reported in the literature. This work does not provide any new genetic funding, but highlight the rareness of the biallelic pathogenic variant in OPA1 gene is, and the case report provided a good and useful description of the clinical process of the proband described in this work. Despite the manuscript was well written and is easy to follow some aspects of the manuscript, especially in the figures, must be addressed to give a clearer version to the reader. In the same way, discussion could be highly improved, and maybe bring in that way some novelty to the current work, by providing more details about the Ser763Valfs*15 variant and the relevance of frameshift mutations in Behr syndrome.

Comments:

  1. In the result/discussion, the authors referred to the 4 cases described by Bonneau et al, but is no refer to the cases reported by Bonifert et al, in which all cases harbour one out of two variants presented in the proband. I think that results and data provide a good amount of information that is not totally discussed in the discussion part. For instance, despite in the results it is indicated that this change p.I437M is not pathogenic itself, which effects does it generate? Is it able to reduce the protein amount?

Likewise, the authors reported that the other variant Ser763Valfs*15 cause the introduction of a premature stop codon resulting is premature protein termination. However, how much functional protein is lost when this variant is present? Has it been estimated? Is it described any splicing study in which the effect of this mutation is in-depth analysed? If any splicing study has been conducted it must be included as well. For the list of 22 cases (21 + proband), this is the only frameshift mutation included. What does implies in the development of Behr syndrome? Is this kind of variant commonly present in non-biallelic cases of Behr syndrome? If so, How does this kind of variant affect to the severity of this disease? I think that addressing all of some of this points in the discussion will be improved considerably it considerably.

Other comments:

  1. ALL figures must be cited in the text the order that is settled. In the manuscript figure 1, is not cited at all. Figure 4 is cited after figure 5 and before figure 6. Re-adjust figure ordering (example, Figure 4 will be figure 5 and the other way around), and ensure all figures were cited along the text in the correct order.
  2. Add a figure representing the family tree of the proband, including proband and his parents together with the genetic background, and the phenotypic observations. This will provide a nice overview of the case report for the readers.
  3. Figure and/or their legends must be completed to provide enough information to the reader:
    • Figure 1 and 5. Legends indicate that optic disc pallor were observed, however, this must be highlighted with an arrow pointed this area. Also, arrow meaning must be included in the legend.
    • Figure 2. The text in the boxes in section (a) and (b) is not discernible, adapt the figure to make this content legible or explain it in the legend. If text is not relevant, remove it from the figure.
    • Figure 3. Like figures 1 and 5 macula must be highlighted somehow and indicate the meaning of the symbol (for instance a box).
    • Figure 4. Legend must be completed by indicating the meaning of the colours showed in graphs and tables, which is the difference between red, yellow and green?
    • Figure 6 and 7. Part (a) must be bigger, part (b) is easy readable, but part (a) is unreadable. As in figure 2 text must be discernible to the reader or explain the meaning in the legend somehow. If text is not relevant, remove it from the figure
  4. In some parts of the manuscript, like the abstract, “biallelic” is written like “bi-allelic”, please correct.
  5. Genes must be written in italics, but no protein. Currently, several genes are written like proteins (for instance genes on lines: 107, 115, 139, 140, 271, etc), but properly written in another part of the manuscript. Please check this along the whole text.

Author Response

Thank you for taking the time to review your manuscript. Attached are the responses to your feedback.

Reviewer 3 Report

The authors report on a novel patient with bi-allelic mutations in OPA1 associated with a severe syndromic form of optic atrophy featuring early onset progressive vision loss, progressive ataxia, seizures, metabolic strokes and mild hearing impairment. The case was followed for a period of 6 years until the age of 8 and the clinical findings – with an emphasis on the ophthalmological work-up – are well described and documented.
The second part of the manuscript includes a literature review on other cases with bi-allelic mutations described in the literature. Given the scarcity of such cases there are no larger case series described so far but rather reports on single or few patients. In as much such an overview is well appreciated. To the best of my knowledge this overview is quite exhaustive except for the cases described by Liao et al. Neurology 88: 131-142 (2017). Besides the tabular listing of the cases the discussion on the bi-allelic OPA1 cases falls somewhat short in the comparative aspect of the clinical findings. Moreover, the preponderance of the Ile437Met variant in the bi-allelic OPA1 cases (12 of the 44 OPA1 alleles in Table 1; note that Ile437Met and Ile382Met most likely denote the very same mutation but annotated based on different reference sequences) should be emphasized.
Table 1 needs a thorough and intensive revision in terms of the uniformity of the variant annotation (best provide alterations both at the mRNA and protein level using the HGVS guidelines), and adjusted to the same reference sequence.
Minor points:
(1) introduction, line 34: the biological functions of OPA1 (in particular its pro fission function) should be described more specifically.
(2) line 41: omit comma
(3) line 56: reference 7 is not a suitable paper to refer to autosomal dominant inherited Behr syndrome. Marelli et al. Brain 134:e169 reported on heterozygous OPA1 mutations in Behr syndrome, although dominant inheritance was not demonstrated in the reported family but only two affected brothers.
(4) lines 124-137: for the argumentation on the pathogenic effect (and effect strength) of the Ile437Met variant it may be helpful to refer to Del Dotto et al. BBA – Molecular Basis of Disease Vol. 1864: 3496-3514 which nicely demonstrated by using functional assays the rather mild effect of this variant.       

Author Response

Thank you for taking the time to review our manuscript. Attached are the responses to your feedback.
